



# Size-resolved aerosol pH over Europe during summer

Maria Zakoura[1,2], Stylianos Kakavas[1,2], Athanasios Nenes[1,3] and Spyros N. Pandis[1,2,4]

[1]Institute of Chemical Engineering Sciences, ICE/FORTH, Patras, Greece,

[2]Department of Chemical Engineering, University of Patras, Patras, Greece,

[3] Ecole Polytechnique Fédérale de Lausanne (EPFL), Switzerland

[4]Department of Chemical Engineering, Carnegie Mellon University, Pittsburgh, USA

*Correspondence to*: Spyros N. Pandis (spyros@chemeng.upatras.gr) and Athanasios Nenes
(athanasios.nenes@epfl.ch).

**Abstract.** The dependence of aerosol acidity on particle size, location and altitude over Europe
during a summertime period is investigated using the hybrid version of aerosol dynamics in the
chemical transport model PMCAMx. The pH changes more with particle size in northern and
southern Europe owing to the enhanced presence of non-volatile cations (Na, Ca, K, Mg) in the
larger particles. Differences of up to 1-4 pH units are predicted between sub- and super-micron
particles, while the average pH of $PM_{1-2.5}$ can be as much as 1 unit higher than that of $PM_1$. Most
aerosol water over continental Europe is associated with $PM_1$, while $PM_{2.5-5}$ and $PM_{5-10}$ dominate
the water content in the marine and coastal areas due to the relatively higher levels of hygroscopic
sea salt. Particles of all sizes become increasingly acidic with altitude (0.5-2 units pH decrease
over 2.5 km) primarily because of the decrease in aerosol liquid water content (driven by humidity
changes) with height. Inorganic nitrate is strongly affected by aerosol pH with the highest average
nitrate levels predicted for the $PM_{2.5-5}$ range and over locations where the pH exceeds 3. Dust tends
to increase aerosol water levels, aerosol pH and nitrate concentrations for all particle sizes. This
effect of dust is quite sensitive to its calcium content. The size-dependent pH differences carry
important implications for pH-sensitive processes in the aerosol.

## 1. Introduction

Acidity is an aerosol property of central importance driving gas-particle partitioning and
heterogeneous chemistry (Pye et al., 2019). pH affects the formation of semi-volatile particulate
matter and the nitrogen cycle by modulating $HNO_3/NO_3^-$ and $NH_3/NH_4^+$ gas-particle partitioning
(Meskhidze et al., 2003; Guo et al., 2017; Nenes et al., 2019). Aerosol acidity can influence pH-





dependent heterogeneous atmospheric processes, like oxidation of $SO_2$ to sulfate, formation of secondary organic aerosol and uptake of $N_2O_5$ on particles (Huang et al., 2011) and also influences aerosol hygroscopicity (Hu et al., 2014). Deposition of acidic particles causes damage on building materials, forests, and aquatic ecosystems (Xue et al., 2011). Aerosol pH can change the solubility of metals, such as iron and copper, which have been linked to aerosol toxicity, and at the same time affects nutrient distributions with impacts on photosynthesis productivity and ocean oxygen levels (Meskhidze et al., 2003; Nenes et al., 2011). Adverse health outcomes have been linked to aerosol acidity, including respiratory diseases (Raizenne et al., 1996), oxidative stress (Fang et al., 2017) and lung and laryngeal cancers (Hsu et al., 2008).

The nitrate partitioning to the aerosol phase is favored when pH exceeds a threshold value (between 1.5 and 3) that depends logarithmically on liquid water content and temperature (Meskhidze et al., 2003; Guo et al., 2016; Nenes et al., 2019). If aerosol pH is high enough (typically above 2.5 to 3), aerosol nitrate formation is favored, as most of the total nitrate formed from $NO_x$ chemistry resides in the aerosol phase. For lower pH values (below 1.5 to 2), formation of aerosol nitrate is not favored and remains in the gas phase as $HNO_3$. Between these pH value limits, a sensitivity window (of 1 to 1.5 pH units) exists in which nitrate can be found either as gas or as aerosol (Vasilakos et al., 2018; Nenes et al., 2019). Atmospheric aerosol has often pH values inside this sensitivity window, for which pH errors could translate to importance biases in aerosol composition (Bougiatioti et al., 2016; Guo et al., 2015, 2017; Vasilakos et al., 2018).

Aerosol acidity and partitioning of semi-volatile species, like nitrate, can be modulated by the presence of soluble inorganic cations of sea salt and mineral dust, such as $Na^+$, $K^+$, $Ca^{2+}$ and $Mg^{2+}$ (Vasilakos et al., 2018). These non-volatile cations (NVCs) tend to reside in the coarse mode of ambient aerosol (sea salt, dust), with much lower concentration in smaller particles (Seinfeld and Pandis, 2006). Chemical transport models tend to overpredict aerosol inorganic nitrate levels in both US and Europe (Yu et al., 2005; Pye et al., 2009; Fountoukis et al., 2011; Tuccella et al., 2012; Heald et al., 2012; Walker et al., 2012; Im et al., 2015; Ciarelli et al., 2016; Zakoura and Pandis, 2018; Zakoura and Pandis, 2019). One of the reasons for these errors is that these models do not simulate properly the aerosol acidity introducing errors in gas-particle partitioning of semi-volatile species, often affecting predictions of inorganic nitrate (Vasilakos et al., 2018).

The aerosol pH has been estimated combining field measurements and aerosol thermodynamic models. Katoshevski et al. (1999) estimated the aerosol pH in the marine boundary layer using the thermodynamic model ISORROPIA and found that it ranged between -0.5 to 9 for particle diameters smaller than 1 µm up to 10 µm. pH was estimated to be 0 to 2 for the




accumulation mode and 2-5 for the coarse mode particles using aerosol and gas phase data collected over the Southern Ocean in combination with the EQUISOLV II model (Fridlind and Jacobson, 2000). Keene et al. (2004) calculated mean pH, ranging from 2.6 to 3.9, for 0.75-25 μm particles based on measurements by an impactor for aerosols and Teflon filters for gases in New England during summer. $PM_{2.5}$ pH was calculated with ISORROPIA II in Beijing during all seasons in 2016-2017 with values ranging from 3.8 to 4.5 (Ding et al., 2019). $PM_{2.5}$ particles are strongly acidic in Hong Kong, China ($[H^+]$=103 nmol $m^{-3}$ for spring of 2001) (Pathak et al., 2003). $PM_{2.5}$ particles during different seasons at an urban site in Guangzhou, China were generally acidic (average $[H^+]$ ~ 70 nmol $m^{-3}$) (Huang et al., 2011). Guo et al. (2015) estimated that $PM_1$ particle pH varied from 0.5 to 2 in the summer and 1 to 3 in the winter in the Southeastern US. $PM_1$ pH was estimated for the northeastern US and its mean value was 0.77 (Guo et al., 2016). $PM_{2.5}$ pH values of 0-2 were estimated combining ISORROPIA II and data collected at a rural southeastern US site during summer 2013 (Weber et al., 2016). Based on impactor measurements in Atlanta, GA during the spring of 2015, Fang et al. (2017) calculated a mean pH value of 3.5 for the coarse mode particles using the ISORROPIA II model. Guo et al. (2017) calculated $PM_1$ and $PM_{2.5}$ pH (equal to 1.9 and 2.7) from measurements during the CalNex study in combination with ISORROPIA II. An average $PM_1$ pH equal to 2.2 was estimated in a rural southeastern US site using ISORROPIA II (Nah et al., 2018). Vasilakos et al. (2018) used the three-dimensional chemical transport model, CMAQ, along with ISORROPIA II, to predict the annual average $PM_{2.5}$ pH over the Eastern US for 2001 and 2011 (pH equal to 1.6 and 2.5, respectively). Bougiatioti et al. (2016) calculated $PM_1$ pH (between -0.97 and 3.75) using ISORROPIA II in the eastern Mediterranean. Squizzato et al. (2013) estimated a mean $PM_{2.5}$ pH value equal to 3.1 over Po Valley, Italy during 2009 based on filter measurements using the E-AIM thermodynamic model. A comprehensive survey of pH studies to date can be found in Pye et al., (2019).

Most of the previous studies focused on the average pH of a particular size range neglecting potential pH variation with particle diameter. There is evidence that pH may vary by as much as 6 units between particle diameters of 0.1 μm to 10 μm (Fang et al., 2017; Ding et al., 2019). The majority of previous work has focused on select locations in the US, Canada and Asia and there is still little information about Europe. Also, there is only one study that links aerosol acidity with altitude (Guo et al., 2016), indicating the need for further investigation.

The aim of our work is to investigate the size-dependent aerosol pH over Europe. For this purpose, the Particulate Matter Comprehensive Air quality Model with extensions, PMCAMx, including the thermodynamic model ISORROPIA II, was used. Europe is particularly interesting,





owing to the large concentration of $NH_3$, nitrate, sulfate and dust across all sizes. The role of dust,

$Ca^{2+}$ and the variation of aerosol pH with altitude are analyzed in detail.

## 2. Model description

PMCAMx (Tsimpidi et al., 2010; Karydis et al., 2010) is based on the CAMx air quality model (Environ, 2003) to simulate the processes of horizontal and vertical advection, horizontal and

vertical diffusion, wet and dry deposition, gas- and aqueous-phase chemistry. A sectional approach is used to dynamically track the evolution of the aerosol mass and composition distribution across 10 size sections covering a diameter range from 40 nm to 40 µm. The aerosol components modeled include sulfate, nitrate, ammonium, sodium, chloride, calcium, potassium, magnesium, other inert crustal material, elemental carbon, water, primary and secondary organic species. The gas-phase

chemical mechanism used in this application is based on the SAPRC mechanism (Carter, 2000; Environ, 2003). The version of SAPRC mechanism used here includes 237 reactions of 91 gases, 18 radicals and 37 aerosol species. The thermodynamics of inorganic species was simulated using the ISORROPIA II model (Fountoukis and Nenes, 2007). Additional details regarding PMCAMx are provided in Fountoukis et al. (2011).

We use the hybrid approach to model inorganic aerosol mass transfer, where for particles with dry diameters less than 1 µm, bulk equilibrium is assumed. For larger particles, the mass transfer to each size section is simulated using the Multicomponent Aerosol Dynamics Model (Pilinis et al., 2000). Trump et al. (2015) used the hybrid approach over Europe to improve the simulation of coarse particle chemistry. They found that $PM_1$ nitrate overprediction in areas with

high sea-salt levels was reduced with the hybrid approach due to the more accurate representation of the interaction of nitric acid and ammonia with coarse mode sea salt. These interactions result in reduction of fine nitrate and increase of nitrate in the coarse mode. Given the importance of pH on the partitioning of nitrate, the hybrid approach is essential for capturing the size-resolved variability of pH.

pH is calculated in this work for particles smaller than 1 µm, 1-2.5 µm, 2.5-5 µm and 5-10 µm, using a molal definition consistent with the $pH_F$ definition of Pye et al. (2019):

$$pH = -\frac{\log\left(1000[H^+]\right)}{[W]},\tag{1}$$

where $[H^+]$ and $[W]$ are the concentrations of particle hydronium ion and particle water in µg m$^{-3}$.



### 3. Model application


PMCAMx was applied over Europe, during the EUCAARI summer intensive campaign in May 2008 for which the model has been evaluated in previous work (Fountoukis et al, 2011). The domain covers a $5400 \times 5832$ km$^2$ region with $36 \times 36$ km grid resolution and 14 vertical layers extending up to 6 km. Inputs to the model include horizontal wind components, vertical diffusivity, temperature,


pressure, water vapor, clouds and rainfall, all generated using the Weather Research and Forecast (WRF) meteorological model (Skamarock et al., 2005). Anthropogenic gas-phase emissions include land emissions from the GEMS dataset (Visschedijk et al., 2007) as well as international shipping emissions. Anthropogenic particulate emissions of organic and elemental carbon were obtained from the EUCAARI Pan-European Carbonaceous Aerosol Inventory (Kumala et al., 2009). Industrial,


domestic, agricultural and traffic emission sources are included in the two inventories. Biogenic emissions were based on MEGAN (Guenther et al., 2006), and sea-salt emission inventories were developed using the approach of O'Dowd et al. (2008). Urban dust emissions were based on the work of Kakavas et al. (2020), assuming that calcium, potassium, magnesium and sodium represented 2.4%, 1.5%, 0.9%, and 1.2% of the emitted mineral dust, respectively. A reliable


Saharan dust emissions inventory was not available; therefore, the African region is excluded from the simulation analysis. More information about the inputs of PMCAMx during the simulated period can be found in Fountoukis et al. (2011) and Kakavas et al. (2020).

Three simulations were performed. The first was the "base case" simulation and included all emissions described above. Two other simulations are carried out and compared with the "base


case" to understand how NVCs in dust affect water uptake and aerosol pH: one where dust lacks any non-volatile soluble cations ("inert dust" simulation) and one where we neglect calcium (the major NVC in dust) from the "base case" simulation. Calcium is unique compared to the other NVCs in that it can react with sulfate ions and form insoluble $CaSO_4$, which precipitates out of the aerosol aqueous phase and remains insoluble under subsaturated conditions - even for metastable aerosol


(Fountoukis and Nenes, 2007). This unique interaction implies that Ca, if present in sufficient amounts, can reduce aerosol sulfate and reduce acidity, but at the same time reduce hygroscopicity that promotes acidity- in a way that is not obvious by just comparing the base case simulation with the "inert dust" simulation. In all simulations, the total dust mass emissions were the same and only its assumed composition varied.




## 4. Results and discussion

### 4.1 Size dependence of aerosol pH

The average ground level pH predictions for different size ranges are presented in Fig. 1. pH is higher over the sea for all particle sizes compared to continental regions due to the presence of sea salt, lower $NH_3$ and the systematically higher RH and liquid water content – all of which act to reduce aerosol acidity. The pH of marine aerosol increases with particle size, with the highest value equal to 4.5 for the 2.5-5 μm and 5-10 μm ranges, as sea salt is emitted mainly at the super-micron

range and is the main aerosol component. Over the continental region, average $PM_1$ pH ranges between 1 to 3 with the highest values in the northern coastal parts of Europe and northern Italy; in these regions, acidity is reduced by the high levels of $NH_3$ present from agriculture and livestock emissions combined with high $NO_x$ and RH levels (Guo et al., 2018; Masiol et al., 2019). $PM_{1-2.5}$ is less acidic with pH values from 1 to 4 over the continental region with the higher values in the

northern coastal areas of Europe (e.g., parts of the United Kingdom). The average pH increases further in the 2.5-5 μm range being equal to 2-3 over the continental regions and reaching values up to 4-4.5 in the northern coastal areas of the United Kingdom, Belgium, France and Poland. Average $PM_{5-10}$ pH slightly decreases over the continental region compared to $PM_{2.5-5}$, especially in central Europe, and the highest pH values (4-4.5) predicted for the same regions. The size-dependence of

pH is stronger in the northern and southern parts of Europe, and weaker in central Europe, in which the average pH is in the 1.5-3 range for all particle sizes. The largest pH changes across size occur for regions where fine-mode aerosol acidity is dominated by the $NH_3$-$SO_4$ system (i.e., relatively lower $NH_3$ levels – so that aerosol nitrate is low; Guo et al., 2018), and the largest sizes contain large amounts of NVCs from sea salt and dust.

Squizzato et al. (2013) calculated $PM_{2.5}$ pH equal to 2.3 and Masiol et al. (2019) equal to 2.2 in the Po Valley, Italy during the summer of 2009 and 2012, respectively. The PMCAMx predictions for this area are in good agreement with these studies, as $PM_{2.5}$ pH is predicted to be equal to 2.4. Guo et al. (2018) estimated that the $PM_{2.5}$ pH was equal to 3.3 in Cabauw, Netherlands during summer 2013, based on measurements and thermodynamic modeling. PMCAMx

underpredicts $PM_{2.5}$ pH by 0.8 units (equal to 2.5) in this area during the summer. Part of this discrepancy could be due to the different periods compared. Bougiatioti et al. (2016) determined through thermodynamic analysis of observations with ISORROPIA-II that the $PM_1$ pH in Finokalia, Crete is equal to 1.3 which agrees within 0.4 units with our predictions (equal to 1.7).

The pH of $PM_{2.5}$ has often been the focus of previous measurement studies, due to the

availability of the corresponding filter samples. However, the pH in the 1-2.5 range can be quite

different from that in the sub-micrometer range (Fang et al., 2017; Ding et al., 2019). This difference may have important implications for aerosol toxicity, metal solubility, nitrate partitioning and other processes. The difference of average ground level aerosol pH predictions between $PM_{1-2.5}$ and $PM_1$ is shown in Fig. 2. The pH of these size ranges can differ up to 1 unit over the continental region.

This difference is even higher over the ocean (up to 1.4 units), owing to the effect of NVCs from sea-salt levels.

Particle water concentrations for the different particle sizes are shown in Fig. 3. $PM_1$ has the most water, compared to the other size fractions over the continental region. The coarse particles in $PM_{2.5-5}$ and $PM_{5-10}$ have the most water over sea owing to sea salt, which is found in higher levels in

these particles, and exhibits the highest hygroscopicity – compared to all other inorganic salts found in aerosol. Water levels for all particle sizes are higher at areas closer to the sea, owing to the relatively high dry aerosol mass concentration combined with the high RH typically associated with the marine environments; in just the 2.5-5 μm size range alone, water content exceeds 20 μg m$^{-3}$.

**4.2 Temporal evolution of pH**

To study the temporal evolution of pH, eight sites (Fig. S1) with different characteristics were selected based on their different type, location and dust/sea salt levels (Table S1). Iza, in Ukraine, has the lowest $PM_1$ pH of all examined locations with a value equal to 0.25 (Fig. S2). The pH is predicted to vary between -0.5 and 1.4 (on an hourly basis) in this region of Eastern Europe that is

characterized by high sulfate levels. Mace Head, on the other hand, has the highest average $PM_1$ pH (1.7) with values ranging between 0.9 and 2.3. Finokalia has the most variable $PM_1$ pH with a range covering 4 units. This site is affected by both relatively dry air masses with continental aerosol characteristics (low pH) and by air masses with relatively high sea-salt and dust levels as well as biomass burning influences (higher pH; Bougiatioti et al., 2016). The distribution of pH values of

the 1-2.5 μm diameter particles moves to higher values (less acidic particles) for all sites compared to the sub-micrometer particles. The pH values of all sites for the 2.5-5 μm range are similar to those in the 5-10 μm range and higher than the fine aerosol pH values.

The pH diurnal profiles for Cabauw, Melpitz, Paris, Finokalia are shown in Fig. 4. These sites were selected based on their different type, location and dust/sea salt levels (Table S1) – and

because ambient pH data is available for both of them (Pye et al., 2019). pH follows the same trend for all particle sizes in each of the four sites. Cabauw is characterized by relatively constant average diurnal pH with slightly higher values early in the morning. The average hourly pH is a lot more variable in Melpitz and Paris, with a peak early in the morning (up to 3.5 for Melpitz and 3.8 for



Paris), and then decreasing values during the day reaching a minimum in the afternoon. The pH
diurnal profile is different in Finokalia, since pH has its peak (up to 3.6) at noon (between 15-16
UTC time) and then starts to decrease. These variations are caused by a variety of factors including
the relative humidity (that is higher during the early morning, leading to higher liquid water content
and higher pH), the temperature (which tends to evaporate nitrate) and mixing height variation
(which in turn tends to affect precursor concentrations).


### 4.3 pH variation with height

All the results presented so far are for the ground level (lowest 50 m). The predicted aerosol water
content for all size ranges decreases with altitude (Fig. S3). This is mainly due to the decrease of the
relative humidity and aerosol concentrations with altitude (Mishra et al., 2015; Wang et al., 2018).
As height increases, pH values for all particle sizes decrease, due to the reduction of aerosol water
per unit mass of dry aerosol, with height (Fig. 5) – which is exclusively an effect of relative
humidity decrease. A secondary effect is that the lower concentration of aerosol tends to drive
partitioning of semi-volatile species (nitrate, ammonium) to the gas phase (Nenes et al., 2019). As a
result, particles of all sizes that are acidic at ground level become more acidic when they move
higher in the atmosphere.

For $PM_1$, in the less acidic areas over Europe the pH decreases from 2-2.5 near the ground to
around 1.5-2 at 2.5 km altitude. For $PM_{1-2.5}$, the reduction is even larger, since the pH values
decrease from 3-3.5 at the ground to 1.5-2.2 at 2.5 km (the larger drop in pH is a result of the
evaporation of nitrate aerosol and the decrease of liquid water content). Similar decreases of 1-1.5
pH units are predicted for the coarse particles in the first 2.5 km of the atmosphere in areas over
land. The predicted decrease in aerosol water content and pH for the super-micrometer particles is
even more pronounced in the marine atmosphere and coastal areas due to the high levels of sea-salt
near the ground. This reduction of pH with altitude is smaller for the $PM_1$ size range in the marine
atmosphere.


### 4.4 Effect of aerosol pH on inorganic nitrate

The highest average nitrate levels (0.7 μg m⁻³) are predicted for the 2.5-5 μm size range (Fig. S4) for
the whole domain.  This size range is characterized by the highest average pH (with a value of 2.63).
Nitrate partitioning to the aerosol phase is favored when the aerosol pH is higher than 2.5 (Guo et
al., 2016; Vasilakos et al., 2018).  At the same time, the mass transfer of the produced nitric acid in
the gas phase is faster for the particles in the 2.5-5 μm range compared to those in the $PM_{5-10}$ range



also contributing to higher concentration. Finally, the removal of the larger particles from the atmosphere is faster adding one more reason for the maximum of the nitrate size distribution.

The size-dependent average nitrate diurnal profiles for Cabauw, Melpitz, Paris, and Finokalia, are shown in Fig. 6. In Cabauw, predicted total nitrate levels start to increase early in the morning, have their peak at noon or afternoon and decrease during the afternoon and early evening. Most of this variation is due to the formation of ammonium nitrate in the $PM_1$ size range. The increase in $PM_1$ nitrate is accompanied by an increase in the ammonium levels (Fig. S5) in this area that is characterized by high ammonia concentrations. The morning increase in the $PM_{5-10}$ nitrate

levels is due to the formation of sodium nitrate and calcium nitrate during daytime. In Cabauw, all super-micron particles have pH above 2.5, favoring the partitioning of nitrate to the aerosol phase forming also ammonium nitrate. In Melpitz, the behavior of nitrate is quite different than in Cabauw due to the differences in the pH behavior (Fig. 4). $PM_1$ nitrate peaks early in the morning with peaks in coarse nitrate a few hours later. The peak in fine nitrate is at the same period as the pH in this

range, while for the coarse particles it is a few hours later due to the delays in mass transfer to the larger particles. Nitrate levels in all size ranges are predicted to decrease during the afternoon with nitrate reaching a minimum in the late afternoon. The behavior of nitrate in the fine and coarse particles in Paris is quite similar as in Melpitz reflecting the similarity in the behavior of pH. Nitrate in all size sections peaks in the early morning and has a minimum in the afternoon. The main

difference in this case is that there is more nitrate in the coarse particles due to the higher predicted levels of dust. In Finokalia, the predicted nitrate increases gradually in all sizes during the morning, reaches its maximum values in the afternoon and then gradually decreases.

## 4.5 Effect of dust on particle pH

The impact of the NVCs from dust on pH can be quantified comparing the results of the simulation in which the dust was assumed to be inert with the base case simulation. Aerosol water levels are higher in all particle sizes for the base case simulation (Fig. S6) compared to the inert dust simulation, as result of the water uptake associated with the NVCs. Dust is predicted to cause an increase of 1.2-2 µg m$^{-3}$ in aerosol water concentration even for the submicrometer particles over

Europe with the highest changes in the northern areas. The water increases for $PM_{1-2.5}$ due to dust varies from 1 to 2.5 µg m$^{-3}$ over continental region. The effect, as expected, is higher for $PM_{2.5-5}$ reaching up to 3 µg m$^{-3}$ in areas like the Po Valley in Italy and even higher for $PM_{5-10}$ ranging between 1 and 6 µg m$^{-3}$. The highest differences in aerosol water levels between the two simulations





are predicted for areas that combine relatively high values of RH and relatively high values of dust
(Fig. S7) during the simulated period.

The predicted aerosol pH is lower in all particle sizes for the inert dust case compared to the base case simulation (Fig. 7). The soluble NVCs in dust tend to increase pH, as due to their lack of volatility they irreversibly neutralize bisulfate ions that are generated by the $NH_3/NH_4^+$ equilibrium, and therefore elevate aerosol pH. NVCs also elevate aerosol water in a way that leads to pH
increase, directly through their hygroscopicity and indirectly, through promoting the condensation of aerosol nitrate (and its associated water content; Guo et al., 2018). $PM_1$ is also affected, even though it contains small amounts of dust, as its pH increase by approximately by 0.1 units or so over parts of continental Europe. The pH increases with diameter by 0.5 units for $PM_{1-2.5}$, 0.8 units for $PM_{2.5-5}$ and 1.4 units for $PM_{5-10}$.

The effect of dust on pH and aerosol water is reflected on the predicted aerosol nitrate. Nitrate in all particle sizes decreases when dust is assumed to be inert (Fig. S8), as the corresponding pH reduction (for cases when pH < 2.5) does not favor the partitioning of nitrate to the aerosol phase. The effect of dust on submicrometer nitrate is negligible in most areas, but there is still an effect in the Netherlands and the surrounding areas (Fig. S8). The dust is predicted to
cause average increases of $PM_{1-2.5}$, $PM_{2.5-5}$ and $PM_{5-10}$ nitrate up to 0.5 µg m$^{-3}$, 1.3 µg m$^{-3}$ and 1.2 µg m$^{-3}$, respectively in parts of northern Europe with higher dust levels and also Italy. This nonlinear impact of relatively minor amounts of NVCs from dust occurs because relatively small changes in aerosol pH, when occurring in the "pH sensitivity window" of nitrate partitioning can lead to large responses in nitrate uptake (Vasilakos et al., 2018).


### 4.5.1 The role of calcium

Predicted aerosol water levels decrease in the absence of calcium compared to the base case simulation (Fig. S9). The increase of aerosol water concentration caused by the calcium ranges between 0.8-1 µg m$^{-3}$ for $PM_1$, 0.8-1.1 µg m$^{-3}$ for $PM_{1-2.5}$, 0.8-1.2 µg m$^{-3}$ for $PM_{2.5-5}$ and 0.7-1.3 µg
m$^{-3}$ for $PM_{5-10}$ over continental Europe. The effect is more significant in the coarse particles where most of the calcium is found. Considering the possible effects calcium can have on soluble sulfate and water uptake, the simulations suggests that the primary effect of calcium is through its action as a soluble ion. If calcium is neglected in the simulation, aerosol pH decreases for all particle sizes compared to the base case simulation (Fig. 8). This decrease varies from 0.2 to 0.3 units for $PM_1$,
0.25-0.4 units for $PM_{1-2.5}$, 0.3-0.5 for $PM_{2.5-5}$ and 0.6-0.8 for $PM_{5-10}$ over continental Europe. The





highest pH differences are predicted for the coarse particles, consistent with that the coarse particles are richest in calcium.

## 5. Conclusions

The size-dependent aerosol pH was simulated over Europe during an early summer period. We find that fine mode aerosol is persistently more acidic than coarse mode particles. The size-dependence of pH is strongest in northern and southern Europe, where the difference can be as large as 4 units between submicron and 10 μm particles. This difference is reduced over continental regions, but can still be as large as 1 pH unit between $PM_1$ and $PM_{1-2.5}$. $PM_1$ has the most water over continental

areas, while $PM_{2.5-5}$ and $PM_{5-10}$ have the most water in the marine and coastal areas.

      Particles of all sizes become increasingly acidic with altitude owing to the reduction of aerosol water levels with height and volatilization of particulate ammonium and nitrate due to dilution. The highest pH decrease between the ground and 2.5 km altitude is 0.5-1 units for $PM_1$, 1.5-2 units for $PM_{1-2.5}$ and $PM_{2.5-5}$, 1.3 units for $PM_{5-10}$. The largest drop in pH is observed for the

$PM_{1-2.5}$ fraction because it coincides with where aerosol nitrate resides most – hence its evaporation with altitude tends to have a larger impact on pH than reductions of liquid water from the RH effect alone.

      The nitrate concentration tends to peak a few hours later than the pH in all examined sites due to the time required for the production of nitric acid and its partitioning to the aerosol phase. If

aerosol pH becomes low enough to impede fine mode nitrate formation, its preferential condensation to larger sizes tends to increase the pH difference across size. The highest average nitrate levels over Europe are predicted for the 2.5-5 μm range for which the average pH is equal to 2.6 during the simulated period.

      Dust causes increases of the aerosol water levels in all particle sizes. The increase in water

levels ranges from 1 to 6 μg m$^{-3}$ with the highest change for $PM_{5-10}$ in parts of northern Europe with relative high concentrations of dust. Dust also causes an increase in aerosol pH for all particle sizes with higher effects in the coarse particles. This effect can be more than 1 pH unit. This increase in pH is accompanied by increases in aerosol nitrate, which can be as large as 2.5 μg m$^{-3}$. This effect of dust is mainly due to its calcium content, suggesting the importance of simulating accurately not

only the dust concentration but also the calcium levels.

      This study clearly shows that aerosol acidity and liquid water content changes considerably across size, location, time and height over Europe. These changes will impact aerosol formation and its response to emissions controls, solubility of aerosol trace metals and deposition. With this



realization, aerosol pH and liquid water content emerge as powerful aerosol state variables (Nenes et
al., 2019) that could help elucidate the complex impacts of aerosol on public health, ecosystems and
climate.

***Acknowledgments.*** This work was supported by the project PyroTRACH (ERC-2016-COG) funded
from H2020-EU.1.1. - Excellent Science - European Research Council (ERC), project ID 726165.

***Code and Data availability.*** Simulation results are available upon request.

***Competing interests.*** The authors declare that they have no conflicts of interest.

***Author contributions.*** MZ, SN and AN conceived and led the study. MZ developed and
implemented the pH calculation scheme in PMCAMx, carried out the simulations and wrote the first
draft of the paper. SK contributed the dust emissions and NVC scheme. MZ, AN and SP were
involved in the scientific interpretation of the simulations. All authors provided feedback on the
analysis approach and extensively commented on the manuscript.

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







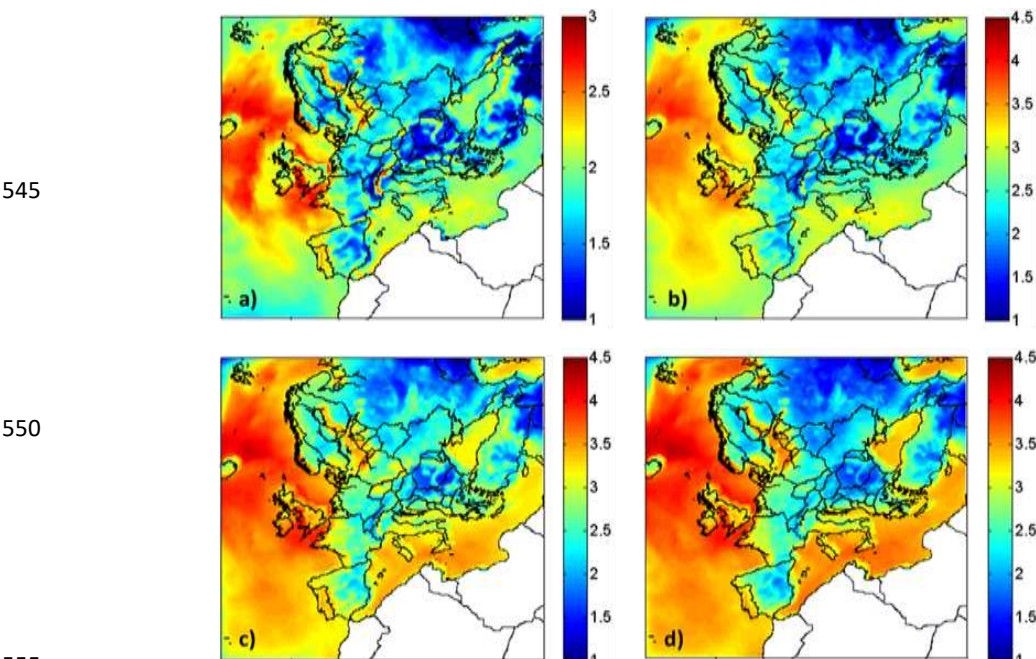

**Figure 1.** Average ground level aerosol pH predictions for **a)** PM$_1$, **b)** PM$_{1-2.5}$, **c)** PM$_{2.5-5}$ and **d)** PM$_{5-10}$ for the base case simulation over Europe during May 2008.












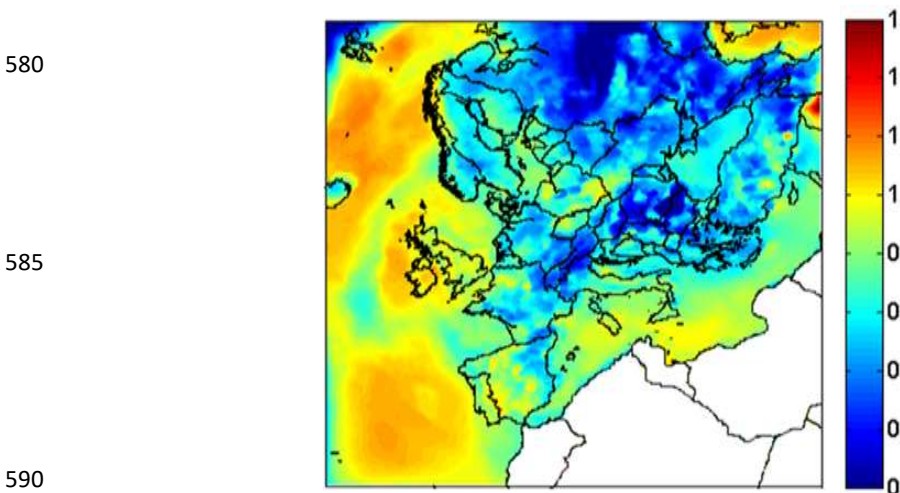

**Figure 2.** Absolute difference of average ground level pH between $PM_{1-2.5}$ and $PM_1$ for the base case simulation during May 2008.








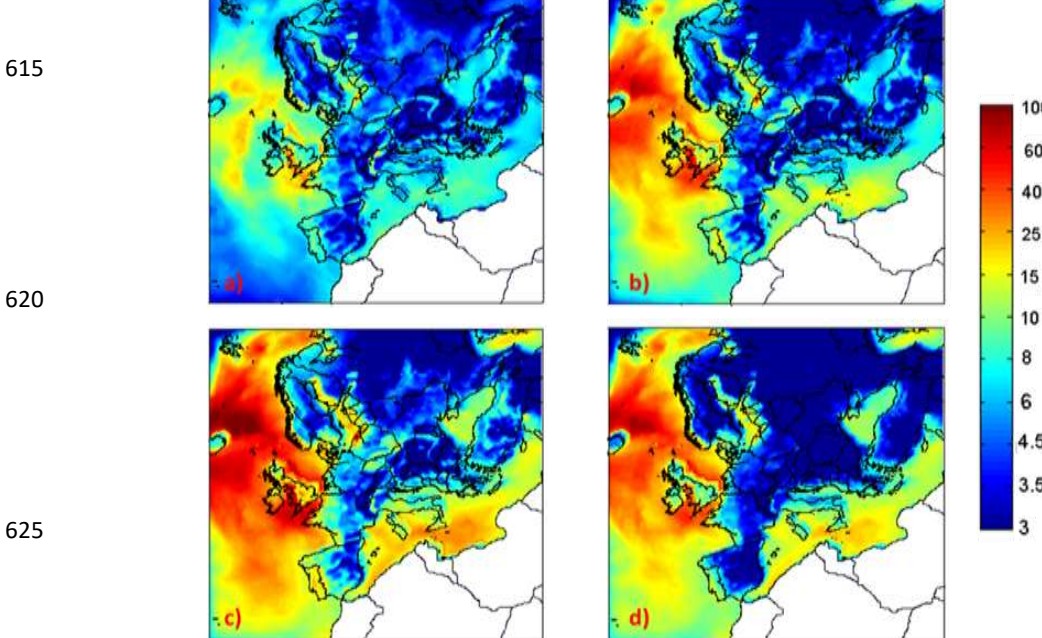




**Figure 3.** Average ground level aerosol water predictions (in μg m$^{-3}$) for **a)** PM$_1$, **b)** PM$_{1-2.5}$, **c)** PM$_{2.5-5}$ and **d)** PM$_{5-10}$ for the base case simulation over Europe during May 2008.






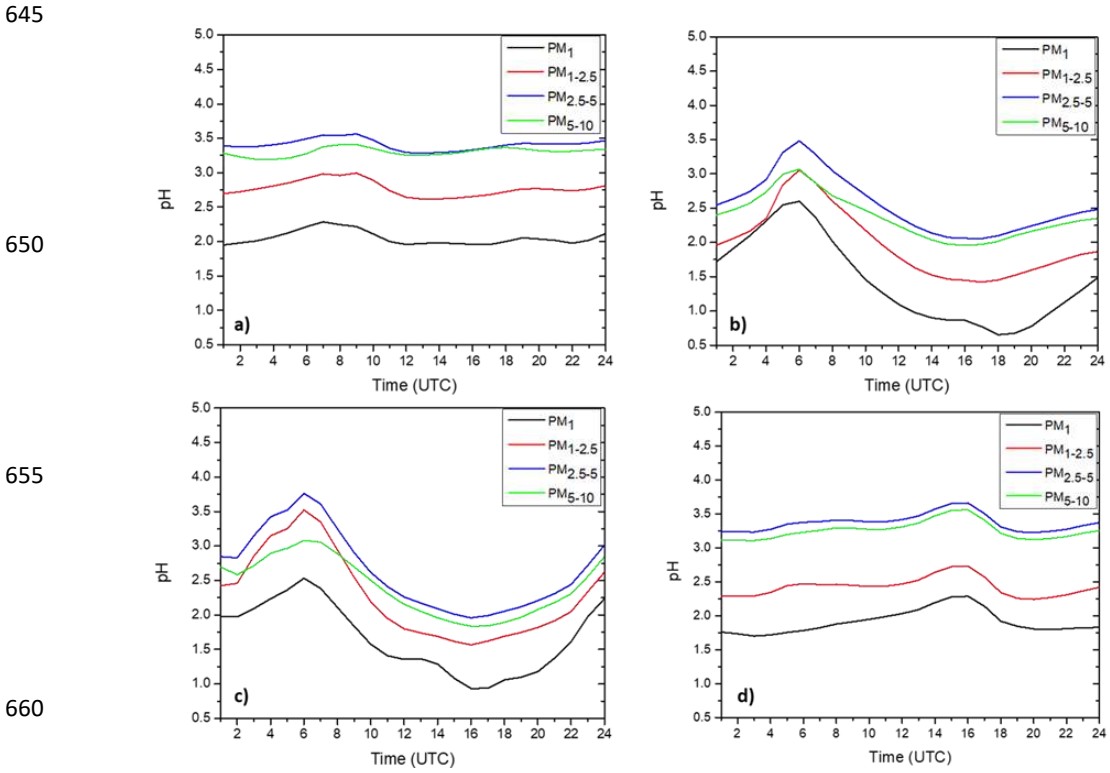




**Figure 4.** Average pH diurnal profiles for **a)** Cabauw, Netherlands, **b)** Melpitz, Germany, **c)** Paris,

France and **d)** Finokalia, Greece for the four particle size ranges for the base case simulation

during May 2008.







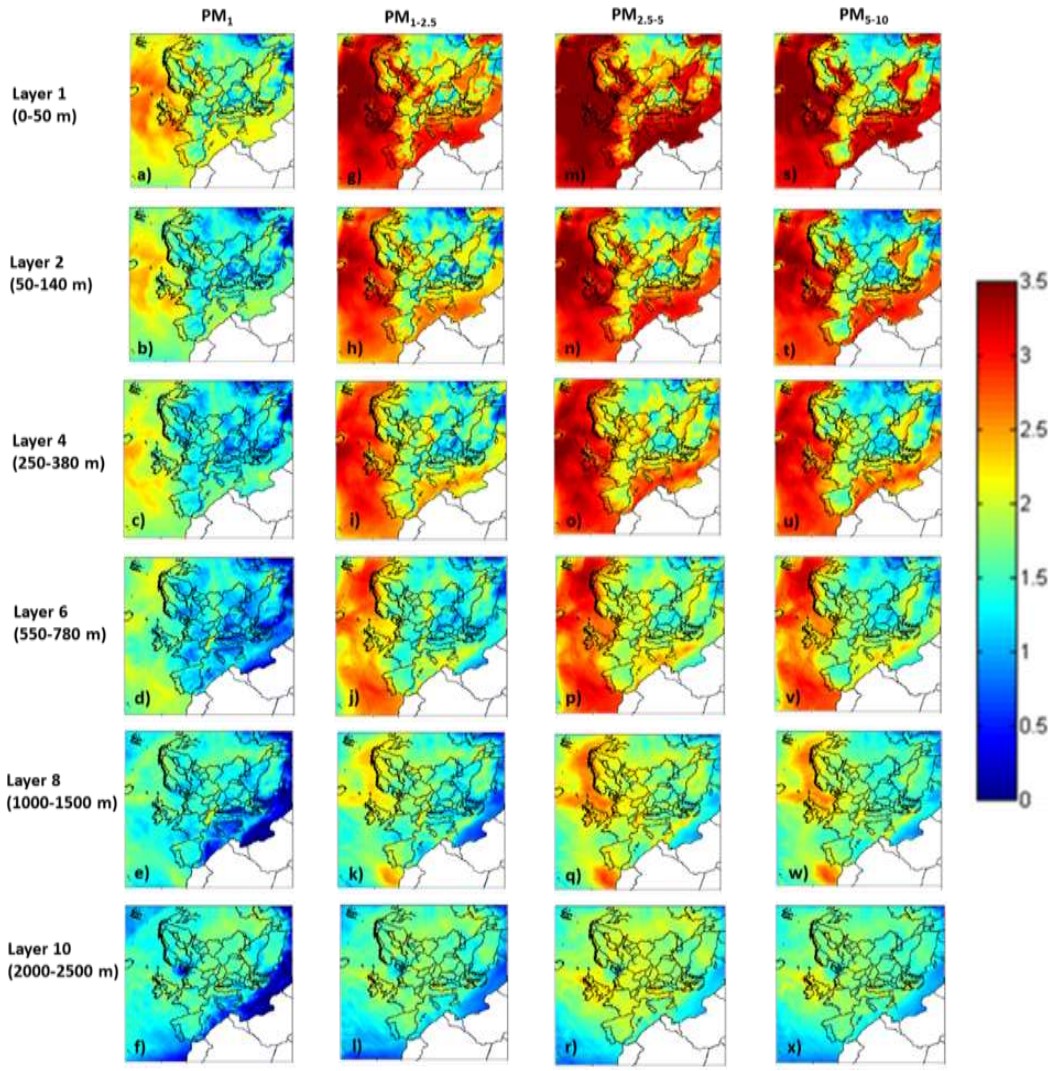

**Figure 5.** Average predicted aerosol pH as a function of size and altitude: **a), b), c), d), e), f)** for

PM$_1$, **g), h), i), j), k), l)** for PM$_{1-2.5}$, **m), n) o), p), q), r)** for PM$_{2.5-5}$, **s), t), u), v), w), x)** for PM$_{5-10}$

at 0-50 m, 50-140 m, 250-380 m, 550-780 m, 1000-1500 m, and 2000-2500 m for the base case

simulation during May 2008.






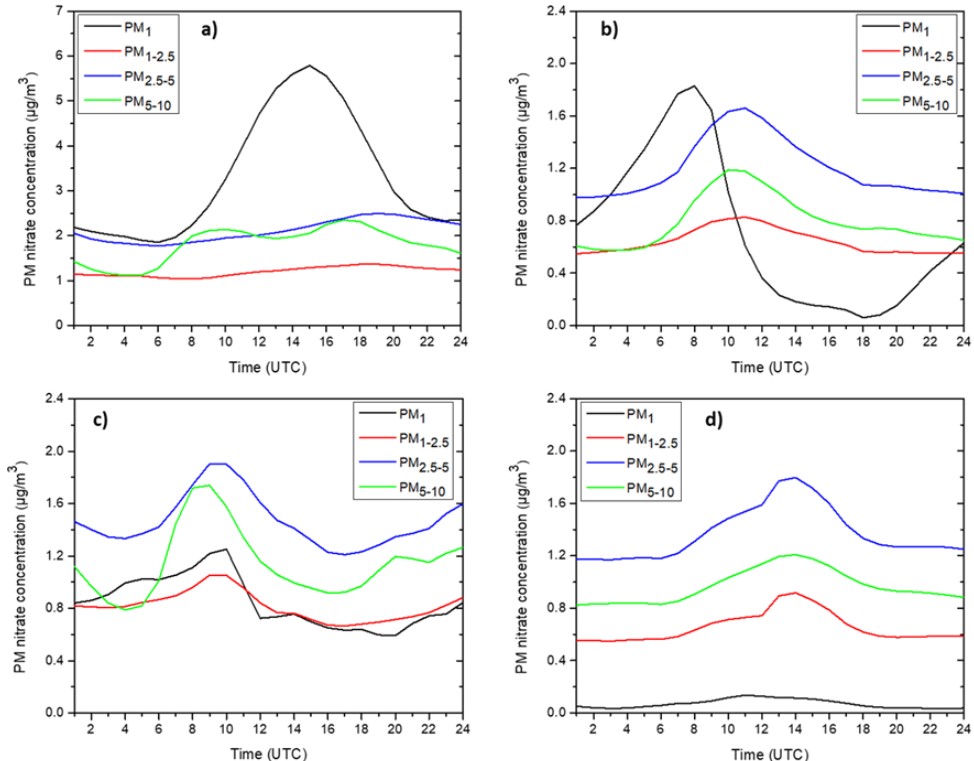


**Figure 6.** $PM_1$, $PM_{1-2.5}$, $PM_{2.5-5}$ and $PM_{5-10}$ nitrate diurnal profiles for **a)** Cabauw, Netherlands, **b)** Melpitz, Germany, **c)** Paris, France and **d)** Finokalia, Greece for the base case simulation during May 2008.











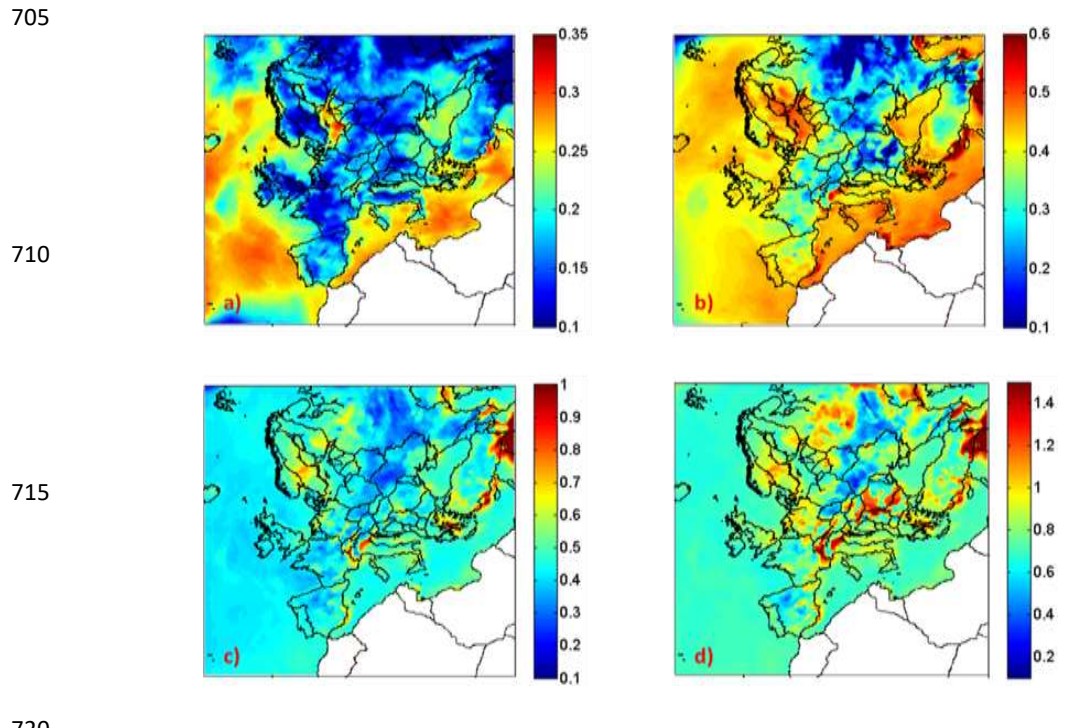

**Figure 7.** Increase of average ground level aerosol pH for **a)** PM$_1$, **b)** PM$_{1-2.5}$, **c)** PM$_{2.5-5}$ and **d)** PM$_{5-10}$ for the base case simulation compared to the inert dust case during May 2008.








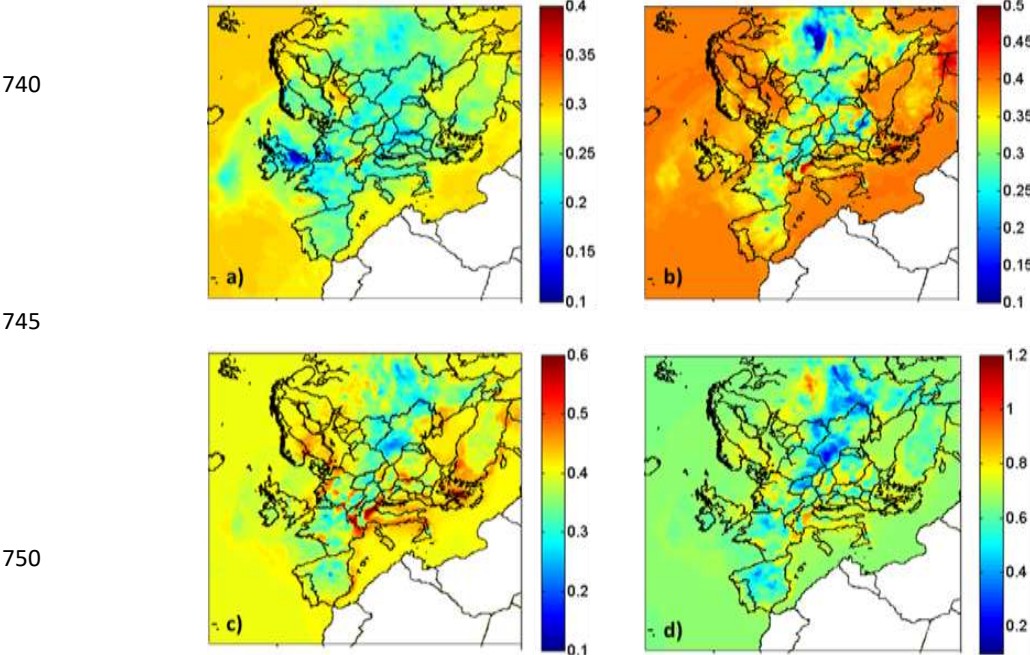





**Figure 8.** Increase of average ground level aerosol pH for **a)** PM$_1$, **b)** PM$_{1-2.5}$, **c)** PM$_{2.5-5}$ and **d)** PM$_{5-10}$ for the base case simulation compared to the case when calcium is neglected during May 2008.