# Peer review of "Size-resolved aerosol pH over Europe during summer"

_Atmospheric Chemistry and Physics, 2019_

## Referee Comment (RC1) · Anonymous Referee #1 · 23 Mar 2020

The paper describes results on size dependent pH in aerosol from the PMCAMx chemical transport model and analyses sensitivity of modeled pH with regard to non-volatile cations. Considering the importance of aerosol pH for aerosol public health, ecosystem and climate effects, the topic is relevant for this journal and this reviewer recommends publication after the following major comment have been addressed.

The paper focuses on May 2008 as the period during which online observations of PM1 composition are available through the EUCAARI intensive campaign at the sites discussed in more detail. While some of the model performance was evaluated in the paper cited (Fountoukis et al., 2011), aspects such as the representation of nitrate diurnal concentration (Figure 6) were not shown before. The diurnal variation of PM1 nitrate shown in figure 6 does not seem to be in accordance with observed diurnal

variation (see e.g. Mensah et al., ACP 2012). Considering the link to aerosol pH, it is critical in the context of this paper to show in detail the performance of the model with respect to aerosol composition. This reviewer therefore requests a detailed analysis of the model ability to simulate particle composition, specifically particulate nitrate. Accounting for the high fraction of organic nitrate in PM1 across Europe (Kiendler-Scharr et al., 2016), this analysis should take into consideration organic nitrate.

Minor points in order of appearance:

The introductory summary of observed aerosol pH (page 2 line 62 – page 3 line 89) is not suitable to provide an overview as is. Switching between reported units (from pH to [H+] in nmol m-3) and listing values without obvious systematic should be avoided and the material structured into e.g. a table or figure to provide an overview.

Page 5 Lines 148ff: When introducing abbreviated names for simulations, apply this to all including the simulation that neglects calcium.

Page 6 Lines 188: Are the two periods for Cabauw (summer 2013 and May 2008) similar in aerosol composition and source, i.e. should one expect the same pH value? If not, why make a comparison and state that PMCAMx under predicts PM2.5 pH by 0.8 units (line 190)?

Page 7 line 225: What does "both" refer to in this context?

Page 8 Line 246: Show comparison of observed and modeled size distribution of nitrate where available.

Page 8 Line 261: If kinetics of mass transfer is critical here, discuss for full size distribution. It is unclear why only the difference between two size ranges is discussed here.

Page 9 Line 264: Compare average diurnal profiles of nitrate with observed data (see also major point above).

[Figure]

References

Fountoukis et al., Atmos. Chem. Phys., 11, 10331 – 10347, 2011

Mensah et al., Atmos. Chem. Phys., 12, 4723–4742, 2012

Kiendler-Scharr et al. (2016), Geophys. Res. Lett., 43, 7735–7744, doi:10.1002/2016GL069239.
* * *

---

## Referee Comment (RC2) · Anonymous Referee #2 · 14 Apr 2020

In this work, Zakoura et al. used a chemical transport model to investigate aerosol acidity over Europe and its variations with particle size, altitude, and time. In addition, they emphasize on the central role of non-volatile mineral cations, notably calcium, on pH calculations and nitrate concentrations. This study is of definite interest to the ACP audience by contributing to one of the least understood atmospheric aerosol properties, the aerosol acidity. The manuscript is very well written, the methodology is scientifically sound, and the presentation is clear. However, I have one major comment regarding the omission of mineral dust emissions from the Sahara Desert. Overall, I recommend this study for publication. Below are a few comments to be considered prior to publication.

Major comment:

[Figure]

1. The authors mention that they have not included the dust emissions from the Sahara Desert. However, it is well known that dust particles from the Sahara can travel towards Europe and influence the air quality over Southern Europe but in some cases as far as the Central Europe. Given the importance of NVCs to aerosol pH calculations and aerosol nitrate formation, as vividly presented in this work, the authors should include the Saharan mineral dust and NVC emissions in their analysis. Mineral dust emissions can be calculated online from the WRF model that has been used here for meteorological inputs. Alternatively, there are available reliable emission inventories to be used offline such as the AEROCOM emission inventory (Dentener, 2006).

Specific comments:

1. Page 2, lines 52-61: A reference to the AeroCom phase III study for aerosol nitrate can also fit in the discussion here (Bian et al., 2017).

2. Page 2 line 64: The value of 9 for aerosol pH seems unrealistically high and certainly not in line with the results of your study.

3. Page 4 line 112: Please change "was" with "is"

4. Page 4 Eq. 1: The [W] needs to be inside the log.

5. Page 4 Eq. 1: Is this the water from ISORROPIA II only or do you also consider the water associated with the organics? If so, please discuss briefly how you calculate the aerosol water associated with the organics and state their hygroscopicity if needed.

6. Page 4 Section 2: Based on the model description, I assume that you don't take into account the impact of organic acids on aerosol pH. Can you briefly discuss the implications of such a simplification?

7. Page 5 line 144: Can you add a reference for these fractions?

8. Page 5 line 154: Do you assume stable or metastable aerosols for the present study?

9. Page 5 line 152-156: Can you explain more here? How the insoluble CaSO4 is removed from the particles? Do you have soluble and insoluble size sections in your model and you calculate the aerosol pH only for the soluble sections or do you have only one well mixed particle for each size section?

10. Page 6 Section 4: I found the map projection used in Fig. 1 and the rest of the manuscript quite confusing, making hard to follow the results. Can you use a different map projection (e.g., Mercator)?

11. Page 6 1st paragraph: Can you state the domain average (or the continental average) aerosol pH for each of the particle sizes?

12. Page 6 1st paragraph: Can you comment on why the tropical Atlantic Ocean in figure 1a looks very acidic with pH values lower than 2? Also, Northern Scotland looks more acidic than the rest of the Great Britain.

13. Page 6 line 167: Why lower NH3 results in higher pH? Do you mean lower sulphate?

14. Page 6 lines 181-184: Very interesting. You can also specifically comment on the Mediterranean Sea where the change of pH between sizes is large. Furthermore, over the Mediterranean, submicron aerosol nitrate is very low and super-micron nitrate very high, corroborating your hypothesis.

15. Page 7 line 225: Do you mean all of them (and not both)?

16. Page 8 lines 231-234: This is very interesting. Can you expand the discussion here? Do these diurnal profiles of aerosol pH correlate with any of the diurnal profiles of the variables stated here (e.g., RH, T, PBL)?

17. Page 8, lines 244-245: This is also very useful result. Can the authors comment if this acidification of aerosols can affect their CCN activity and/or the pH of the formed cloud droplets?

18. Page 8 Section 4.4: Why there is a nitrate-free zone between North and South Europe in Figure S4? I would expect that NOx and NH3 emissions are everywhere in central Europe. Furthermore, it looks like you have more nitrates over the oceans than over land.

19. Page 8 Line 260: Do you mean the mass transfer to the aerosol?

20. Page 10 lines 296-304: Why the effect over oceans is so large in Fig. 7? What is the composition of sea salt emissions? Have you changed their composition here as well?

21. Page 10 lines 296-304: The impact of NVCs on aerosol pH and nitrate is quite impressive, given that you only use urban dust emissions. This emphasizes the need to include Saharan emissions as well.

22. Page 10 line 303: Do you mean by up to 0.5 units?

23. Page 10 line 308: This is not the case here. Over the northern coast of continental Europe and Southern England, the impact of NVCs on nitrate concentrations is significant despite the fact that the impact on pH is negligible. Why submicron aerosol nitrate has such a strong increase (almost twofold) in the presence of NVCs?

24. Page 10 Section 4.5.1: Similar to NVCs, The impact of calcium on the pH all over the oceans is very strong. Does your sea salt contain any Ca? Can you comment why pH increases almost uniformly even over the remote oceanic locations of your domain?

References:

Bian, H. S., Chin, M., Hauglustaine, D. A., Schulz, M., Myhre, G., Bauer, S. E., Lund, M. T., Karydis, V. A., Kucsera, T. L., Pan, X. H., Pozzer, A., Skeie, R. B., Steenrod, S. D., Sudo, K., Tsigaridis, K., Tsimpidi, A. P., and Tsyro, S. G.: Investigation of global particulate nitrate from the AeroCom phase III experiment, Atmospheric Chemistry and Physics, 17, 12911-12940, 10.5194/acp-17-12911-2017, 2017.

Dentener, F., Kinne, S., Bond, T., Boucher, O., Cofala, J., Generoso, S., Ginoux, P., Gong, S., Hoelzemann, J. J., Ito, A., Marelli, L., Penner, J. E., Putaud, J. P., Textor, C., Schulz, M., van der Werf, G. R., and Wilson, J.: Emissions of primary aerosol and precursor gases in the years 2000 and 1750 prescribed data-sets for AeroCom, Atmos. Chem. Phys., 6, 4321-4344, 2006.

---

## Author Comment (AC1) · 27 May 2020

**General comment**

**(1)** *The paper describes results on size dependent pH in aerosol from the PMCAMx chemical transport model and analyses sensitivity of modeled pH with regard to non-volatile cations. Considering the importance of aerosol pH for aerosol public health, ecosystem and climate effects, the topic is relevant for this journal and this reviewer recommends publication after the following major comment have been addressed.*

We appreciate the positive assessment of our work and the careful review of our work by the reviewer. We have tried to address all comments of the reviewer and to improve the paper accordingly. Our responses (in regular font) and the corresponding changes

in the manuscript follow each comment of the reviewer (in italics).

**Specific comments**

**(2)** *The paper focuses on May 2008 as the period during which online observations of PM1 composition are available through the EUCAARI intensive campaign at the sites discussed in more detail. While some of the model performance was evaluated in the paper cited (Fountoukis et al., 2011), aspects such as the representation of nitrate diurnal concentration (Figure 6) were not shown before. The diurnal variation of $PM_1$ nitrate shown in Figure 6 does not seem to be in accordance with observed diurnal variation (see e.g. Mensah et al., ACP 2012). Considering the link to aerosol pH, it is critical in the context of this paper to show in detail the performance of the model with respect to aerosol composition. This reviewer therefore requests a detailed analysis of the model ability to simulate particle composition, specifically particulate nitrate. Accounting for the high fraction of organic nitrate in $PM_1$ across Europe (Kiendler-Scharr et al., 2016), this analysis should take into consideration organic nitrate.*

We have followed the suggestion of the reviewer and added an analysis of the model performance for nitrate. Overall, the results are quite similar to the previous applications of PMCAMx for this period (including the Fountoukis et al., 2011 study). The model does capture the observed nitrate diurnal variation in the corresponding sites. There was an unfortunate error in Figure 6a with the nitrate diurnal variation shown for Cabauw and we thank the reviewer for noticing it. The data shown corresponded to another area in the modeling domain. This figure has been corrected. The model does predict, consistent with the observations in Cabauw, that the fine nitrate peaked on average in the early morning. We have also added a discussion of the organonitrate fraction in the revised paper. The model predicts only inorganic nitrate, therefore the comparisons with the AMS results should be based also on the inorganic nitrate.

*(3) The introductory summary of observed aerosol pH (page 2 line 62 – page 3 line 89)*
*is not suitable to provide an overview as is. Switching between reported units (from pH to [H$^+$] in nmol m$^{-3}$) and listing values without obvious systematic should be avoided and the material structured into e.g. a table or figure to provide an overview.*

We have rewritten this section of the introduction focusing on the pH and the existing information about its size dependence. We also discuss briefly the recently published review of Pye et al. (2019) that includes a detailed survey of such measurements. This review includes tables so we do not repeat them in the present study.

**(4)** *Page 5 Lines 148ff: When introducing abbreviated names for simulations, apply this to all including the simulation that neglects calcium.*

We have followed the reviewer's suggestion and now use the abbreviated name "no calcium" for the simulation where we neglect calcium.

**(5)** *Page 6 Lines 188: Are the two periods for Cabauw (summer 2013 and May 2008) similar in aerosol composition and source, i.e. should one expect the same pH value? If not, why make a comparison and state that PMCAMx under predicts PM$_{2.5}$ pH by 0.8 units (line 190)?*

The aerosol pH can be sensitive to small changes in composition but also to meteorology (especially relative humidity). These differ seasonally so differences from year to year are expected that can be easily as much as 0.5 pH units. So while it is expected that the early summer period (May 2008) in our study and the summer 2013 period studied by Guo et al. (2018) do not have major differences concerning the emissions and meteorology, it is dangerous to reach quantitative conclusions comparing the corresponding pH values. Our goal here was to investigate if our model predicts reasonable pH values compared to the values calculated based on measurements and thermodynamic models for similar periods. We have added a brief discussion of this point in the revised manuscript.

**(6)** *Page 7 line 225: What does "both" refer to in this context?*

We replaced "both" with "all" in the revised manuscript.

**(7)** *Page 8 Line 246: Show comparison of observed and modeled size distribution of nitrate where available.*

Please note that this section in the paper focuses on the vertical distribution of nitrate. The overall agreement of PMCAMx predictions with the airborne data is encouraging. The ability of the model to reproduce the high time resolution airborne measurements at multiple altitudes and locations is quite similar to its ability to capture the ground level (hourly) observations. A comparison of the average vertical profiles for the flights of EUCAARI is shown in Figure 8c of Fountoukis et al. (2011). A brief discussion has been added. We have also added a paragraph discussing the predicted nitrate size distributions and their comparison with the available measurements.

**(8)** *Page 8 Line 261: If kinetics of mass transfer is critical here, discuss for full size distribution. It is unclear why only the difference between two size ranges is discussed here.*

PMCAMx uses a sectional scheme for the description of the aerosol size-composition distribution. There is one size bin extending from 2.5 to 5 $\mu$m and one from 5 to 10 $\mu$m. This is the reason that the discussion in this point focuses on the differences between the two size ranges that cover the coarse mode above 2.5 $\mu$m. There is no other information to show about the size distribution in this size range. This discussion is trying to address the factors affecting the nitrate distribution in the coarse particles. We have added a reminder to the reader at this point about the size resolution used by the model to avoid confusion.

**(9)** *Page 9 Line 264: Compare average diurnal profiles of nitrate with observed data (see also major point above).*

We have followed the reviewer's suggestions and added comparisons of the diurnal variation of the predicted nitrate with the available observations. Overall the model is successful in capturing the corresponding patterns.

---

## Author Comment (AC2) · 27 May 2020

**General comment**

**(1)** *In this work, Zakoura et al. used a chemical transport model to investigate aerosol acidity over Europe and its variations with particle size, altitude, and time. In addition, they emphasize on the central role of non-volatile mineral cations, notably calcium, on pH calculations and nitrate concentrations. This study is of definite interest to the ACP audience by contributing to one of the least understood atmospheric aerosol properties, the aerosol acidity. The manuscript is very well written, the methodology is scientifically sound, and the presentation is clear. However, I have one major comment regarding the omission of mineral dust emissions from the Sahara Desert. Overall, I*

[Figure]

*recommend this study for publication. Below are a few comments to be considered prior to publication.*

We appreciate the positive assessment of our work and the careful review of our work by the reviewer. Our responses (in regular font) and the corresponding changes in the manuscript follow each comment of the reviewer (in italics).

**Major comment**

**(2)** *The authors mention that they have not included the dust emissions from the Sahara Desert. However, it is well known that dust particles from the Sahara can travel towards Europe and influence the air quality over Southern Europe but in some cases as far as the Central Europe. Given the importance of NVCs to aerosol pH calculations and aerosol nitrate formation, as vividly presented in this work, the authors should include the Saharan mineral dust and NVC emissions in their analysis. Mineral dust emissions can be calculated online from the WRF model that has been used here for meteorological inputs. Alternatively, there are available reliable emission inventories to be used offline such as the AEROCOM emission inventory (Dentener, 2006).*

We agree with the reviewer about the interesting features of Saharan dust episodes and their impacts on both air quality and pH of aerosol over Southern Europe and even further north. We made the decision to focus first on periods during which the impact of Saharan dust in Europe is minimal. This is the majority of the time. Our plan is to investigate dust events in the next step of this work. We do need to address not only the dust emissions and transport but also the other rather uncertain anthropogenic emissions from Northern Africa. This is now explained in the revised paper.

**Specific comments**

**(3)** *Page 2, lines 52-61: A reference to the AeroCom phase III study for aerosol nitrate can also fit in the discussion here (Bian et al., 2017).*

The recommended reference has added to the revised manuscript.

**(4)** *Page 2 line 64: The value of 9 for aerosol pH seems unrealistically high and certainly not in line with the results of your study.*

We have rephrased this sentence given that the major point here is that Katoshevski et al. (1999) predicted that the pH of the submicrometer marine aerosol is several units lower than that of the supermicrometer particles.

**(5)** *Page 4 line 112: Please change "was" with "is".*

Done.

**(6)** *Page 4 Eq. 1: The [W] needs to be inside the log.*

Done.

**(7)** *Page 4 Eq. 1: Is this the water from ISORROPIA II only or do you also consider the water associated with the organics? If so, please discuss briefly how you calculate the aerosol water associated with the organics and state their hygroscopicity if needed.*

In Eq. 1, [W] represents the concentration of particle water calculated from ISORROPIA II. The water associated with organics is neglected in our study, since most of the time water concentrations associated with organics are about 1/10 of those associated with inorganic aerosol components (Bougiatioti et al., 2016). As a result, the error would be small in our first effort to simulate aerosol pH across particle size. The water associated with organics will be calculated and used for pH calculation in future study. This simplification is now explicitly stated at this point in the paper.

**(8)** *Page 4 Section 2: Based on the model description, I assume that you don't take*

*into account the impact of organic acids on aerosol pH. Can you briefly discuss the implications of such a simplification?*

This is correct, the effect of the organic acids on aerosol pH is not considered in this study. A brief discussion of the literature regarding this effect has been added to the revised paper.

**(9)** *Page 5 line 144: Can you add a reference for these fractions?*

The requested references have been added in the revised manuscript.

**(10)** *Page 5 line 154: Do you assume stable or metastable aerosols for the present study?*

We assumed stable aerosols for the present study. This is now explained in the paper.

**(9)** *Page 5 line 152-156: Can you explain more here? How the insoluble CaSO4 is removed from the particles? Do you have soluble and insoluble size sections in your model and you calculate the aerosol pH only for the soluble sections or do you have only one well mixed particle for each size section?*

The model assumes that each size section in internally mixed, therefore all particles in that size range has the same composition. However, particles in the same section can contain both insoluble material and soluble. Therefore, the insoluble CaSO4 is treated as such by ISORROPIA-II. This information has also been added to the paper.

**(10)** *Page 6 Section 4: I found the map projection used in Fig. 1 and the rest of the manuscript quite confusing, making hard to follow the results. Can you use a different map projection (e.g., Mercator)?*

The map projection used in Fig. 1 as well as in the rest of the manuscript is the polar stereographic map projection that is actually used by PMCAMx in these simulations

to cover most of Europe with fewer computational cells. Using other projections for the results is possible, but it results in empty areas in the graphs (those outside of the modeling domain) or in not showing all the modeling domain. For this reason, we prefer to maintain these maps.

**(11)** *Page 6 1st paragraph: Can you state the domain average (or the continental average) aerosol pH for each of the particle sizes?*

The continental average pH is 1.7 for $PM_1$, 2.2 for $PM_{1-2.5}$, 2.6 for $PM_{2.5-5}$ and 2.5 for $PM_{5-10}$. This information is included in the revised manuscript.

**(12)** *Page 6 1st paragraph: Can you comment on why the tropical Atlantic Ocean in Figure 1a looks very acidic with pH values lower than 2? Also, Northern Scotland looks more acidic than the rest of the Great Britain.*

Figure 1a shows the average ground aerosol pH predictions for $PM_1$. Particles of this size range are predicted to contain relatively little sea-salt and significant concentrations of sulfates resulting in relatively low pH. Obviously the situation is quite different for supermicrometer particles in these regions. Northern Scotland is predicted to have lower pH values compared to the rest of Great Britain for all size ranges during the simulated. This happens because particles across all sizes have lower water content (Fig. 3) in this area. Also, the predicted dust concentrations are lower than the rest of Great Britain (Fig. S7). The lower aerosol water content and lower dust concentrations lead to more acidic particles in Northern Scotland for all size ranges. These observations and explanations have been added to the paper.

**(13)** *Page 6 line 167: Why lower NH3 results in higher pH? Do you mean lower sulphate?*

This is a good point. We have deleted these two words. While this is the case, it does not contribute to higher pH but to lower pH values.

**(14)** *Page 6 lines 181-184: Very interesting. You can also specifically comment on the Mediterranean Sea where the change of pH between sizes is large. Furthermore, over the Mediterranean, submicron aerosol nitrate is very low and super-micron nitrate very high, corroborating your hypothesis.*

We have followed the reviewer's suggestion and added a discussion of the behavior of the aerosol pH, fine and coarse nitrate over the Mediterranean Sea. We have added a discussion of the existing measurements of fine and coarse nitrate in the area that are consistent with the model predictions.

**(15)** *Page 7 line 225: Do you mean all of them (and not both)?*

We mean most of them. The correction has been made in the revised manuscript.

**(16)** *Page 8 lines 231-234: This is very interesting. Can you expand the discussion here? Do these diurnal profiles of aerosol pH correlate with any of the diurnal profiles of the variables stated here (e.g., RH, T, PBL)?*

The aerosol pH diurnal profiles in the four examined sites of our study follow the same pattern as the corresponding RH diurnal profiles. RH values are higher during the early morning, leading to higher liquid water content and higher pH values, for all sites except for Finokalia. RH and pH profiles follow each other in Finokalia too, but they peak at noon and then they start to decrease. This discussion has been added to the paper and the corresponding figure has been added to the supplementary information.

**(17)** *Page 8, lines 244-245: This is also very useful result. Can the authors comment if this acidification of aerosols can affect their CCN activity and/or the pH of the formed cloud droplets?*

This is an interesting question that has a rather complicated answer. The lowering of the pH can drive nitric acid from the particles to the gas phase and lower the CCN

activity of the particles. However, this nitric acid will be available for recondensation as the particle water is increasing during the activation process and may cancel this effect. We will try to look into this issue in future work simulating the detailed activation of such particle populations.

**(18)** *Page 8 Section 4.4: Why there is a nitrate-free zone between North and South Europe in Figure S4? I would expect that NOx and NH3 emissions are everywhere in central Europe. Furthermore, it looks like you have more nitrates over the oceans than over land.*

Please note that the model predicts significant concentrations of nitrate in Central Europe, it just predicts even more in parts of Northern and in Southern Europe. The area around Belgium, Netherlands and the UK had high ammonia levels and together with the high NOx emissions and photochemistry, according to the model, resulted in high levels of both high and coarse nitrates. In parts of Southern Europe, the relatively high sea-salt resulted in relatively high levels of coarse nitrate. Central Europe had all the components but not in such high levels, so there was a minimum in predicted nitrate in that area. An explanation of this interesting behavior has been added to the paper.

**(19)** *Page 8 Line 260: Do you mean the mass transfer to the aerosol?*

Yes. The correction has been made in the revised manuscript.

**(20)** *Page 10 lines 296-304: Why the effect over oceans is so large in Fig. 7? What is the composition of sea salt emissions? Have you changed their composition here as well?*

The initial sea-salt composition has remained constant in this sensitivity test. Of course, the composition changes during the simulation as sulfuric and nitric acid may condense on the sea-salt particles, chloride may evaporate, etc. The effect depicted here is due

to the dust both from emissions inside the modeling domain but also from the boundary conditions (long range transport from outside the domain). The model is predicting that the marine areas, having already a higher pH compared to the continental areas are more sensitive to the dust non-volatile cations. This is now explained in the revised paper.

**(21)** *Page 10 lines 296-304: The impact of NVCs on aerosol pH and nitrate is quite impressive, given that you only use urban dust emissions. This emphasizes the need to include Saharan emissions as well.*

Please note that some Sahara dust emissions are included in the model indirectly as boundary conditions in the south. However, these are constant and lead to moderate transport of dust to Europe and not to so called Sahara dust episodes. We agree with the reviewer that these are clearly important and we will focus on them in future work. This is now explained in the revised text.

**(22)** *Page 10 line 303: Do you mean by up to 0.5 units?*

No. The predicted pH for $PM_1$ when dust is present increases by 0.1 units or so over continental Europe (Fig. 7a). This is clarified in the revised text.

**(23)** *Page 10 line 308: This is not the case here. Over the northern coast of continental Europe and Southern England, the impact of NVCs on nitrate concentrations is significant despite the fact that the impact on pH is negligible. Why submicron aerosol nitrate has such a strong increase (almost twofold) in the presence of NVCs?*

This is a good point and it deserves additional discussion. The pH in this region is in the 2-3 range in which there is significant nitrate in both the gas and particulate phases. These values are inside the "sensitivity window" that we described in the introduction and nitrate is quite sensitive to changes in the cations. The NVCs cause

a non-negligible change in the pH (around 0.2 units) and cause an increase of the fine nitrate of 20-30 percent (not twofold). Some of these changes are not entirely clear due to the color schemes used in the different maps. A brief discussion of the submicron nitrate in this area has been added to the paper.

**(24)** *Page 10 Section 4.5.1: Similar to NVCs, The impact of calcium on the pH all over the oceans is very strong. Does your sea salt contain any Ca? Can you comment why pH increases almost uniformly even over the remote oceanic locations of your domain?*

The calcium over the marine areas in these simulations is due to transport both from the continental areas but also from outside the domain (through the dust boundary conditions). These boundary conditions for dust are now shown in the Supplementary Information (Table S2). The emitted sea salt in our simulations has zero calcium. These once more stress the importance of dust for the pH of the marine atmosphere. This point has been added to the paper.